# When Higher Education Meets Sustainable Development of Rural Areas: Lessons Learned from a Community–University Partnership

Maria Cecilia Mancini, Filippo Arfini and Marianna Guareschi *

Department of Economics and Management, University of Parma, 43125 Parma, Italy;
mariacecilia.mancini@unipr.it (M.C.M.); filippo.arfini@unipr.it (F.A.)
* Correspondence: marianna.guareschi@unipr.it

**Abstract:** Sustainable development in mountain areas faces numerous socioenvironmental and economic challenges that public institutions have sought to address for decades. The EU is increasingly demanding Higher Education Institutions be more socially relevant and responsible in addressing the needs of these often-underserved communities. To this end, one central principle of the Erasmus+ Project 2014–2020, also consolidated by the Programme 2021–2027, has been to enhance the development, transfer, and implementation of innovative practices fostering social engagement at the local and international levels. The paper describes a case study developed in the "Socially Engaged Universities—(SEU)" project, co-funded by the Erasmus+ Programme 2014–2020, which aimed to strengthen the cooperation between universities and local communities. A local foundation and the University of Parma co-piloted a project to strengthen the short supply chains of a group of farmers in the Italian Appennines. The partnership identified factors that fostered and hampered the effectiveness of community–university partnerships. We propose some final recommendations to ensure that sustainable rural development fully leverages university resources. This includes higher education teaching and research programmes tailored toward the needs of the local community.

**Keywords:** short food supply chains; community–university partnership (CUP's); rural development; Socially Engaged Universities (SEUs); Erasmus+; social constraints

## 1. Introduction

The debate on the contribution to sustainable development became central among Higher Education Institutions (HEIs) in 2015 when the United Nations set the 2030 Agenda and 17 Sustainable Development Goals (SDGs), oriented at reaching, preserving, and enhancing the global public goods. Besides the direct involvement in SDG 4 on "quality education", universities can be drivers for achieving several SDGs through knowledge production, innovation, and human formation (Chankseliani and McCowan 2021). (Dryjanska et al. 2022) considered quality education to be, at the same time, "a goal and a precursor to and an outcome to other goals". The SDG Fund identifies the main ways for universities to contribute to the 2030 Agenda for Sustainable Development: expanding human capital with an SDG perspective, conducting research on SGDs topics, implementing the agenda, transferring knowledge, and building the tools that the SDGs will require. In asserting education as a prerequisite to sustainable development, the United Nations Economic Commission for Europe (UNECE) "Strategy for Education for Sustainable Development" (ESD) supports participatory teachings to motivate and empower learners to change their behaviour and take action for sustainable development, thus promoting the implementation of the acquired knowledge for practical use in the solution of ecological and social problems (Nasibulina 2015).

The commitment of HEIs toward ESD is made explicit in the Global University Leaders Forum (GULF), comprising 29 world-leading universities. In 2021, the participants agreed

on five key elements: implementing the concept of sustainable development, improving expertise on sustainable development, supporting scientific research in response to global challenges, fostering collaboration with international partners to provide innovative solutions, and constructive transnational cooperation on specific issues. In Italy, the Conference of Italian University Rectors (CRUI) promoted the Network of Universities for Sustainable Development (RUS) in 2015, the first experience of coordination and sharing among all Italian universities committed to environmental sustainability and social responsibility. The network's main purpose is to disseminate the culture and good practices of sustainability, both within and outside the universities (at the local, regional, national, and international levels), and to contribute to the achievement of the SDGs.

In such a framework, an important concept supported by ESD is social innovation. It is defined as a process that provides solutions to systemic and complex social issues not fulfilled by the existing actors (market players or governmental bodies) to increase people's living standards and welfare through the introduction of novel approaches (Paunescu 2014; Chow et al. 2019).

According to this definition, social innovation produces public goods, introducing solutions to societal needs. Ravazzoli and Valero (2020) considered social innovation an essential tool for SDGs fulfilment by contributing to the sustainability, resilience, and inclusiveness of cities and communities. According to Păunescu et al. (2022), HEIs play an essential role in social innovation production by partnering with other institutions, companies, and society and building trust among the actors involved, as well as engaging in activities such as the coproduction of knowledge with end users, community outreach, policy advice, and using research results outside the academy.

Therefore, HEIs can contribute to SDG fulfilment through the so-called "first mission"—teaching sustainable development topics, "second mission"—research on sustainable development, and "third mission"—engagement in creating innovative strategies within communities for sustainable development. The third mission can be defined as all activities of universities that serve a social interest of development in any sphere of civil society, address non-academic partners, and are not exclusively assigned to the university's core tasks of research and teaching (Henke et al. 2016; Roessler et al. 2015; for an overview Henke et al. 2017). According to Morcellini and Martino (2005), the traditional academic model based on the separation between universities and institutions has been replaced by a model aiming to increase the relationship and exchange between universities, the market, and society, the latter encouraging universities to produce and transfer knowledge for economic and social purposes (Martino 2018). This is consistent with the growing calls for universities to address the SDG challenges and to be more socially relevant and responsible by addressing the needs of society, both locally and globally.

The social engagement of HEIs is a priority in European higher education policies. Indeed, the European Commission, in the framework of the Gothenburg Social Summit vision for 2025, is supporting a network of European universities that favours international exchanges of students, academic staff, and researchers, as well as structured cooperation between HEIs and other public institutions in different countries. To this end, the EU Erasmus+ programme supports education, training, youth, and sports in Europe by promoting synergies and cross-fertilization, removing artificial boundaries between the various actions and project formats, fostering new ideas, attracting new actors from the world of work and civil society, and stimulating new forms of cooperation. Specifically, the Erasmus+ Programme 2014–2020, in key action 2, wants "cooperation for innovation and the exchange of good practices", supports the strategic partnerships in the fields of education, training, and youth aimed at promoting innovation and the exchange of experience and know-how between different types of organizations involved in education, training, and youth or other relevant fields. In such a context, one priority is supporting the social engagement of HEIs and promoting students' intercultural and civic competencies. These principles have been confirmed by the EU Erasmus + programme 2021–2027, which benefits from a budget of EUR 26.2 billion, nearly double the funding compared to its predecessor programme

(2014–2020). The programme supports priorities and activities set out in the European Education Area, Digital Education Action Plan, and the European Skills Agenda. The 2021–2027 programme places a strong focus on social inclusion, green and digital transitions, and the promotion of young people's participation in civic society. This latter priority is addressing the citizens' limited participation in its democratic processes and overcoming the difficulties in students actively engaging and participating in their communities. To this end, nonformal learning is acknowledged as an efficient tool to enhance citizens' social and civic engagement through formal or nonformal learning activities, in addition to formal education.

A conceptual definition of HEI social engagements is offered by the term "community–university partnerships (CUPs)", described as "institutionalized partnerships between universities and their local communities that seek to facilitate teaching, research, and volunteering opportunities for mutual benefit. They involve a long-term commitment to work with community organizations and citizens to collectively address pressing social issues, combining the expertise and resources of the university with the knowledge, resources, and energy of local communities" (Harney and Wills 2017, p. 7).

Harney and Wills (2017) saw CUPs as having two main functions: (i) to broker relationships between people and organizations in the community and people within the university, including students, faculty, and staff and (ii) to facilitate CUP activity. CUPs support the development and maintenance of community engagement activities by ensuring effective communication, developing mutual understanding, and helping to design and find funding for projects. Broadly speaking, CUPs engage in three types of activities in their attempts to work collectively towards addressing social issues: community based-research, understood as a collaborative enterprise between researchers (faculty and students) and community members, which validates multiple sources of knowledge and has a goal of social action and change (Stoecker 2003); a knowledge exchange, which combines the academic and professional expertise of staff within the university with local knowledge held by the community to create innovative solutions to social issues; and student and staff volunteering, which implies that volunteers can develop new and support existing projects in the community to expand the capacity of community organizations to deliver services and meet community needs.

In this framework, the Socially Engaged Universities project (SEU), funded by the Erasmus+ program 2014–2020, aimed to share the experience and know-how of the relationships between European universities and their civil societies and use this as the basis for a series of pilot projects in each partner city (Exeter, Magdeburg, Ghent, Parma, and Delft). Each city–university partnership has brought its own unique relationship and cultural context to the project exploring the expectations that communities have regarding the universities' contributions and innovative engagement approaches and ways for communities to become embedded into research and teaching.

The SEU project aims to contribute to increasing the understanding of the conditions required to support mutually beneficial, sustainable community–university partnerships. This knowledge can add value to developing innovative and feasible policy tools at the university and European level for supporting, monitoring, and assessing the community engagement of universities. By adding to the evidence-based community–university partnerships in different societal contexts, SEU supports policy-makers in encouraging universities to become more engaged.

In this paper, the research question is how to promote effective CUPs by identifying its main hindering and facilitating factors. To this end, the authors analyse a pilot project that the University of Parma was called to co-design and implement within the SEU project. Specifically, UNIPR joined an existing project, "Parma, a mountain of quality", implemented in its first stage by a local foundation to strengthen the short supply chains of farmers placed in the mountainous areas in the Italian Appennines. The next sections are structured as follows: in Section 2, the methodology of analysis is drawn by adapting Drahota et al.'s (2016) list of facilitating and hindering factors that impact CUP effectiveness.

Section 3 describes the project by providing the socioeconomic and geographical contexts, the aim, and the partners involved, as well as the main activities. Section 4 analyses the project by identifying the main facilitating and hindering factors that have fostered or hampered the mutual benefits from CUPs. Section 5 offers some final remarks addressing recommendations to promote effective CUPs.

## 2. Materials and Methods

The project has been analysed through a slightly adapted framework offered by Drahota et al. (2016) that classifies the factors that facilitate and hinder CUPs. They identified twelve facilitating and eleven hindering factors that influence the collaboration process between academic and non-academic partners.

Facilitating and hindering factors refer to the three main spheres that impact the quality of the CUP: the relational aspects, being the key elements in the relation between the parties to build a shared vision, mission, and objectives nourished by a sense of trust; the governance, defined by the leadership, the selection of partners, and a clear definition of their role in the CUP; and the perception of the CUP outcomes by the partners themselves.

Drahota et al. (2016) found that the most frequently reported facilitating factors are trust and respect among partners and a shared vision, goals, and mission. Similarly, Pellecchia et al. (2018) claimed that building and maintaining trust is the foundation of any working partnership between academic and community partners. Building trust can entail, for example, "showing up" for the community and supporting community activities (Collins et al. 2018). Ensuring that all participants are heard and valued within the partnership is also a vital part of gaining trust (Harney and Wills 2017), this being achieved via frequent and clear communication of the expectations and the needs of each of the participating sectors at the onset and throughout the partnership as expectations and needs change (Jernigan et al. 2015). They explain that trust is vital, because universities have long been perceived as "outside" their surrounding communities, operating in their interests rather than in the community's interests. They comment that, from an investment perspective, it is perhaps wiser for universities to invest in building relationships and trust within specific places rather than with more disjointed partners, as trust acts as a resource to support future work without the need to build new relationships from scratch. Place-based approaches can also help generate tangible impacts that meaningfully address complex social issues and demonstrate the worth of the partnership. They argue that a place-based approach to problem-solving is more likely to have the impact that universities and communities are looking for due to its ability to approach social problems in a more holistic and multi-faceted way than is possible via issue-based work (Harney and Wills 2017).

Strictly related to the relational facilitating factors are the governance factors centred on the good quality of the leadership that implies the ability to clearly define differentiated roles and a good selection of partners, as well as the efficient management of the CUP activities, thus including well-structured meetings and effective conflict resolutions.

As far as the outputs, mutuality is seen as a key characteristic by proponents of community engagement, as all CUPs are underpinned by strong relationships of reciprocity between people within universities and communities (Harney and Wills 2017). A clear perception of such mutuality has to reach all components so that the efforts and strengths are addressed for an effective CUP design and management. Indeed, it is crucial for the research question of each partnership to be grounded in the interests and needs of the community (Collins et al. 2018) and for there to be meaningful benefits, results, and impacts for both parties to avoid inconsistent participation of members of the partnership (Mirza et al. 2018). In Martikke's study, adhering to the principles of coproduction was seen as the gold standard of partnership working. Although challenges remain in practice, there is a recognition that a coproduction has several benefits, such as: ensuring that CUPs have mutual benefits; enhancing the impact and quality of research; generating appropriate and ethically sensitive research approaches; enabling practice-relevant outputs; and securing

buy-in and ownership by both partners and their respective stakeholders (i.e., organization, service users, and wider community) (Martikke et al. 2015).

Meanwhile, the most frequently reported hindering factor reported in the Drahota review is an excessive time commitment, which can result in partners' dropping out, followed by unclear roles and functions of partners and excessive funding pressures or control struggles. The former entails that many or all of the partners do not know what their role in the group is supposed to be, or they are not even assigned any roles, thus frustrating their willingness to take an active part in the project; the latter entails partners struggling over the control of funding or external pressures from funding sources that interfere with the decisions, CUP outcomes, or its progress.

However, Gomez et al. (2021) argued that relational factors can most of all undermine the collaborative process, especially during the initiation and development of CUPs, and find that discrepancies often emerges between community and academic partners to the extent that researchers report a greater concern about the organizational process factors, such as differential benefits from participation in the CUP and partner selection, than do community providers. Moreover, half of the community providers considered the "lack of a common language and shared terms" to hinder the collaborative group process, whereas no researchers selected this factor as a hindrance.

Mirza et al. (2018) discussed how to foster CUPs and develop specific strategies to address the facilitating and hindering factors identified by Drahota et al. (2016). The authors argued that the contextual understanding of how barriers to partnerships can be negotiated and addressed within specific projects would provide greater insight into the best practices (Mirza et al. 2018). They suggested that, with careful planning and communication, barriers to community–academic collaborations can be addressed in ways that benefit all parties and facilitate its continued use within marginalized communities.

The complete list of facilitating and hindering factors is available in Tables 1 and 2.

**Table 1.** Facilitating factors of CUPs.

| Category | Facilitating Factors | Definition |
|---|---|---|
| Relational | Trust between partners | Partners have faith in the honesty, integrity, reliability, and/or competence of one another. Partners are comfortable with sharing because they believe that the sensitive information they provide will remain in the group. |
| | Respect among partners | Partners honour and value one another's opinions. Partners are careful to ensure that each member is able to share his or her beliefs. |
| | Good relationship between partners | Partners work well together, form a cohesive group and strong reciprocal relationships, get along well, and like one another. |
| | Effective and/or frequent communication | Partners engage in ongoing communication that is open and respectful. Communication encompasses personal and professional matters. |
| | Shared vision, goals, and/or mission | Partners share the same identified vision or values. Partners identify the same goals or mission for CUP. |
| Governance | Well-structured meetings | Meetings are held with satisfactory or effective frequency. The logistics of the meetings facilitate productivity, satisfaction, effectiveness, partnership, opportunities to interact, etc. (e.g., food available, formality/lack of formality at meetings). The style of the meeting is satisfactory (e.g., face-to-face, telephone, web-based). |
| | Clearly differentiated roles/functions of partners | Each partner has a specific role in the group that contributes to its progress. CUP has a specific group structure with different roles for different partners. |
| | Good quality of leadership | The leader is a person with strong and experienced leadership skills. The leader is open, listens, and takes suggestions into consideration. The leader cares about the members of the group. |
| | Effective conflict resolution | Conflicts are discussed and resolved openly by partners. The team develops as it deals with problems, tensions, and frustrations. |
| | Good selection of partners | The "right" people are selected to be a part of the collaborative group. The personality characteristics of partners contribute to the CUP's success. |
| Output | Positive community impact | Partners perceive the group as having/will have a positive impact on the community. |
| | Mutual benefit for all partners | All partners benefit from the group's progress. Benefits may be different, but all receive some. |

Source: authors' elaboration from Drahota et al. (2016).

**Table 2.** Hindering factors of CUPs.

| Category | Hindering Factors | Definition |
|---|---|---|
| Relational | Mistrust among partners | Partners do not have faith in one another's honesty, integrity, reliability, and/or competence. Partners are uncomfortable sharing because they believe that the sensitive information that they provide in the CUP will not remain in the group. |
| | Partners do not value one another's opinions | Partners make no effort to ensure that each member can share his or her beliefs. |
| | Poor communication among partners | CUP has limited or unclear methods of communication. Partners have trouble maintaining communication. |
| | Lack of shared vision, goals, and/or mission | The CUP has an unclear or undefined vision, goals, values, or mission. Partners have different agendas/visions for the CUP. |
| | Differing expectations of partners | Struggles emerge because not all members expect the same structure, procedures, and/or outcomes. |
| Governance | Unclear roles and/or functions of partners | Many or all of the partners do not know what their role in the group is supposed to be. Partners are not assigned any roles and therefore do not know how they can best contribute to the CUP. |
| | Inconsistent partner participation or membership | Partners attend meetings inconsistently. CUP membership is inconsistent, with attrition or turnover in partnering agencies/organizations or individuals. |
| | Lack of common language or shared terms among partners | Partners lack common terms or definitions related to the topic of interest or work of the CUP. Partners lack a shared understanding of the terms used. |
| | Excessive time commitment | Partners leave the group, want to leave the group, or the CUP does not work well because the time the partners have to spend collaborating is too long. |
| | High burden of activities/tasks | Some, many, or all members are dissatisfied with the amount of work they have to do in order to sustain the CUP. Partners are dissatisfied because the tasks they have to complete are boring, expensive, not meaningful, or otherwise upsetting. |
| | Excessive funding pressures or control struggles | Partners struggle over control of funding. CUP experiences external pressures from funding sources related to decisions, CUP outcomes, or its progress. |

Source: authors' elaboration from Drahota et al. (2016).

## 3. Results

The Province of Parma, located in the Emilia Romagna Region, is characterized by a unique geographical pattern that includes flat lands, hills, and mountains. Whereas intensive agricultural models prevail in the former, extensive, and organic techniques and protected areas, regional and national parks are prevalent in the latter. Some mountainous agri-food products hold the Designation of Origin (PDO) and Protected Geographical Indication (PGI) (e.g., Parma Ham, Parmigiano Reggiano cheese, and mushrooms from Borgotaro) labels, some others are traditional products from ancient varieties of plants and animals for which there is a growing interest in the market (Mancini 2012; Mancini and Consiglieri 2016). Most of the farms are family-run, and despite some engagement of public institutions, they do not have developed consolidated channels for locally marketing their products. Since 2018, a local foundation has launched a project to promote the sustainable development of the short supply chains (Mancini and Arfini 2018; Arfini et al. 2019) of sixty family-managed farms by developing a strategy to make their products known to the urban consumers of the nearest city, Parma. A quality scheme and a label called "Parma, mountains of quality" were created to communicate the place-based quality of the products, but when the time came to define the strategy to reach the urban market, the foundation called for a partnership with the local university, the University of Parma.

### 3.1. Partners, Roles, and the Aim of the Pilot Project

The analysis of the "Parma, mountains of quality" project entails the definition of the beneficiaries and their aims, the main partners involved, and their roles.

The main beneficiaries of the project are the producers of the fragile areas of the Appennines, whereas the other parties of the project are two local institutions, namely the local foundation and the University of Parma. Fondazione Borri is a local foundation that has among its statutory aims to pursue public purposes in the fields of study, research, and training and is particularly committed to promoting cooperation and the equitable development of the Parma area and the University of Parma (UNIPR), a medium-sized

public institution that, for the sake of "Parma, mountains of quality" purposes, has involved some researchers (three senior researchers and one research fellow) of the Department of Economics and Management. These two institutions hold close relationships due to their joint involvement in previous projects on local development.

The main aims of each partner—producers, the foundation, and UNIPR—were defined. Whereas the producers' and foundation's aims can be identified, respectively, as commercial benefits and supporting the local development of a fragile area, UNIPR's aim was twofold. Indeed, UNIPR fulfilled both one of its three fundamental functions, namely, the third mission, but also met the requirements of the SEU project—that is, to explore the mutual synergies emerging from the partnership, as well as the main facilitating and hindering factors that fostered or hampered the fruitful and mutual cooperation.

As project promoter, the foundation played a facilitating role between the producers and UNIPR, whereas UNIPR supported the foundation with academic skills and resources (staff, students, and equipment).

*3.2. Pilot Project Activities*

In January 2020, the foundation and UNIPR had some introductory meetings where the project was described together with its aim, state of the art, and challenges.

One UNIPR researcher was appointed to coordinate the project, acting as a pivot between the students and the farmers (=producers).

A kick-off in-person meeting with all farmers was supposed to take place, but it was cancelled due to the COVID-19 pandemic. The foundation and UNIPR co-designed and implemented the project activities as follows:

A.     General agreement

To institutionalize their shared principles and long-lasting partnership, the foundation and UNIPR signed a general agreement. The agreement was aimed at a " ... collaborative relationship between the Parties, in which the research and teaching activities of the University and the activities of the Foundation are mutually integrated and coordinated, concerning the agri-food systems in the Parma area, with particular attention to (i) sustainable agriculture: in particular organic farming; (ii) the recovery, preservation, protection, and enhancement of animal and plant genetic resources (with particular regard to local agro-biodiversity); (iii) the characterization and valorization of fresh and processed agri-food products of the territory, as well as of agricultural by-products through marketing activities; (iv) the support activities for the management of agri-food enterprises in the Parma area".

B.     Individual meetings with the key farmers

In February 2020, UNIPR conducted open interviews with the key farmers identified among those who expressed more interest and availability in proactively taking part in the project. Their activity, motivations and principles, their farm's history, and main perceived difficulties in improving their business were investigated. The interviews were video-recorded, edited, and subtitled by the Center for the Performing Arts and Professions (CAPAS) of UNIPR.

C.     On-field research

The master's degree in Economics and Management of sustainable food systems at the University of Parma is a two-year degree that combines classes in agri-food and environmental economics, food technology, and sociology. Two students attending the master's classes served their internship within the project and prepared a questionnaire to be submitted to all farmers joining the project (60 in total). Tutored by UNIPR researchers, they gathered both quantitative and qualitative information. The former included the type of products and production volumes; production techniques; distribution channels; and adopted quality labels (organic, PDO, PGI, etc.), whereas the latter included the farm and family background and the main problems and obstacles faced in running the farm activity.

They were also asked to provide pictures of the products, the activity, and the family. The estimated time to complete each of the two parts was 30 min. However, the quantitative part sometimes required extra time for additional explanations, whereas the duration of the storytelling depended on the producer's willingness to talk to the interviewer.

Due to the pandemic, telephone instead of face-to-face interviews were conducted. Twenty-seven farmers were available for interviews in the period August–October 2020.

The data were elaborated and analysed by the same students. The farms were mapped and classified as per farm groups by location, size, product category, labels, and distribution channels. The research showed that 60% of the farms are less than 10 hectares, and the urban market (i.e., Parma) is 30–70 km away for 70% of the farms. The main product categories are honey and marmalade, cereals and cereal-based products, eggs and meat, and dairy products. The most frequent labels are "organic" and "mountain product". Nine farmers use two or more labels, eleven farmers one label, and seven farmers do not use any label. The distribution channels are on-farm shops, restaurants, online stores, and some stores in near villages. Small farms find it difficult to participate in farmers' markets or sell in Solidarity Purchasing Groups. Generally, the most relevant problems concern product marketing (logistics and customer loyalty).

The qualitative data analysis was based on a thematic content analysis, and the information was organized per content: farm/family histories, interests in farm activities, and main perceived problems.

The activities of the internship and the main findings of the analysis were described in the two students' master's degree theses defended, respectively, in November 2020 and March 2021.

A third master's degree student interviewed five producers to receive their views on potential benefits deriving from UNIPR participation.

A fourth trainee student of the master's degree carried out a market analysis by contacting twelve stores and making a price analysis of the leading competitors' products to define a price able to catch consumers' interest and producers' expectations.

All stores were available for the "Parma, mountains of quality" products. The findings were used by the farmers to set prices and by the student to write her master's degree thesis, defended in November 2021.

The students also participated in the design of a website, "ParmaLocalFood" (www. parmalocalfood.unipr.it accessed on 6 June 2022), which was developed with a twofold purpose: on the one hand, to give visibility to mountain farmers and products and, on the other hand, to act as an interactive laboratory between the farmers and the students. The final version of the site will feature a fact sheet for each farmer, providing the products, pictures collected during the survey, and the places and hours where the products are available to consumers. The website also includes a section where students upload their research, videos, interviews, papers, and articles concerning sustainable value chains and quality products in the Province of Parma. A dedicated blog connects the farmers and the students, the former submitting questions, issues, and problems that the students are called to in-class debate for proposals and solutions. The website is designed by the University E-Learning and Multimedia Service Center of UNIPR.

D. Exchanges with third HEIs

Alongside the activities, UNIPR had exchanges of ideas with two Italian Universities (Politecnico di Milano, POLIMI, and Università di Macerata, UNIMC) that have long experience with the third mission initiatives. POLIMI, a member of "ApeNEt" (the Italian Network of Universities and Research Institutions for Public Engagement), developed the "Polisocial" Programme to promote and encourage new multidisciplinary research and didactic activities oriented to human and social development to renew the university's method of research and teaching and increase a responsible attitude and to develop skills, competencies, and new values in future generations of professionals and citizens. UNIMC performed a ten-year process of public engagement working with local stakeholders, the community, and students to support rural development and entrepreneurs networking.

POLIMI and UNIMC experiences were presented in the SEU international meeting held in Ghent in November 2019. As well as strengthening the relations among the three universities, UNIMC invited two UNIPR students to participate in the V International Student Competition on Place Branding and Mediterranean Diet that was supposed to be held from 5 to 10 May 2020 in Fermo (Marche), but it was then cancelled due to the COVID-19 emergency. Nevertheless, the collaboration between the universities continues.

E.     The final SEU meeting

In the final SEU meeting (October 2021), the SEU project partners and the local partners (the foundation, the key farmers, and UNIMC for the UNIPR project) discussed the main hindering and facilitating factors of all pilot projects in four parallel sessions: how to enhance the trust between local stakeholders and the university, how to solve societal challenges through intersectoral and multi-stakeholder collaborations, building long term, sustainable, mutually beneficial relationships between the university and the community, and taking a place-based approach to community engagement.

*3.3. The Tool Kit for Dissemination Activities*

One of the main SEU outputs is a toolkit that offers a package of knowledge to be shared with other institutions interested in undertaking a CUP by providing case studies, infographics, and videos/films.

The SEU partners agreed that the toolkit had to be designed for universities and all potential partners of CUPs by bringing together highlights and lessons from the projects and providing a range of easily accessible "tools".

The UNIPR toolbox included:

- a video that tells the story of the territory through the farmers' eyes and footage of their farms but also interviews with a foundation representative and UNIPR researchers and students, discussing the difficulties and strengths of the project;
- a recorded webinar where product and producer mapping, together with the video, were presented to the farmers, the farmers' Union representatives, local institutions (Municipality and Emilia Romagna Region), and Parma citizens;
- a podcast in which a UNIPR researcher and a foundation representative were interviewed, discussing the main lessons learned;
- a flowchart prepared by a UNIPR master student that describes the different phases of the project, the materials used, the outcomes, and outputs;
- the presentation of the "ParmaLocalFood" (www.parmalocalfood.unipr.it accessed on 8 June 2022) website.

## 4. Discussion: Analysis of the Main Facilitating and Hindering Factors

The influence of facilitating and hindering factors on the collaborative process in the "Parma, a mountain of quality" project was explored alongside the project, specifically in the video interviews, the survey, the additional interviews, and the final SEU meeting.

One of the main recurring themes concerns the role of trustful relationships as a facilitating factor. In the project "Parma, mountains of quality", the discussion on trustful relationships takes place over several levels of analysis. The first one involves the trust relationship between UNIPR and the foundation.

These two institutions agree that trust takes time to develop and needs to be nourished, thus being worth it for institutions to invest in building relationships within specific places rather than with more disjointed partners, similar to Harney and Wills's findings (2017). Indeed, UNIPR and the foundation relationship is based on mutual trust built up over years of commitment to joint projects, thus assuring the awareness to share the same vision, goals, and mission. To some extent, the same applies to the relationship between the foundation and some farmers—more specifically, those farmers identified as key farmers, with whom collaboration began in 2018 based on shared principles. A different story concerns the relationship between UNIPR and the farmers. Aside from some extra project

contacts held by one of the researchers with some key farmers, the trust between UNIPR and the farmers had to be built up. To this end, the researchers faced a relational gap, but they also had to overcome a widespread mistrust perceived by many farmers towards the university and public institutions, grounded on several projects that ended up with no consolidated advantages for them. These represent major problems that must be overcome by reassigning the university to the role of addressing the communities' issues and concerns, as well as helping serve their interests (Hurtado et al. 2012). Indeed, it is a matter of fact that public trust in universities appears to be decreasing, and universities are considered relatively self-centred in the eyes of the public (Van Vught 2021). The fourth level concerns the relationship between farmers themselves that appears to be lacking trust, at least among some of them, and it could not be improved due to the social distancing enforced by the COVID-19 sanitary emergency that prevented the adoption of participatory trust-building tools, such as the organisation of in-person meetings. Indeed, a facilitating factor, such as communication, has been particularly impacted by the pandemic. The kick-off meeting, which was supposed to gather all farmers in person, had to be cancelled and, except for the in-person video recordings with the key farmers and a few interviews, all the activities had to be converted into online initiatives. Social distancing not only negatively impacted the relations between the farmers and UNIPR but also between the farmers themselves. The lack of a real closeness acted as a hindering factor and has likely been a reason for the low answer rate: only 27 producers (45% of the total farmers involved) took part in the phone/Skype survey. The exceptional times when this project took place showed how important real closeness is to overcome mistrust among the participants. It also represented an opportunity to remark on the "digital divide" issue that still excludes those who hardly use technological tools—typically the older generation unable to participate in online meetings and other online initiatives and unwilling to learn how to use video call software.

The lack of real closeness is a critical factor that negatively impacted the sense of belonging and ultimately enlarged the partners' participation. This behaviour very much reflects a major problem in the context of marginalised areas where the geographical distance between producers enhances individualist behaviour and prevents effective associationism.

The project governance has also been impacted by the pandemic. Many farmers complained about the excessive time commitment in taking part in the survey, although the main part of the interviews was completed in September and October 2020, when the most intense working period for farmers declined. Whereas time commitment may have played a role in not taking part in the survey, UNIPR and the foundation think that the lack of involvement might be due to other reasons. First, different partners' expectations may have acted as hindering factors, thus compromising the full success of the initiative. Indeed, the foundation and UNIPR aimed to promote the development of the rural area, thus spending much time reaching as many farmers as possible. This can be perceived as a project weakness by farmers who are quickly receptive to project principles or simply interested in developing commercial partnerships as quickly as possible. Here stands another hindering factor: the time needed by public institutions such as the university to develop activities, thus clashing against farmers' expectations of fast solutions. The dilemma is that HEIs hold skills, resources, and equipment, but it takes time to make them available to the community for several bureaucratic reasons, thus frustrating both researchers' work and farmers' expectations. The third aspect is related to the role assignments. In the case of "Parma, mountains of quality", a professional would help manage those aspects that from the UNIPR and foundation functions, such as the products' delivery management to urban stores. Indeed, although the producers themselves carry out delivery, planning is needed whereby each producer periodically is responsible for the delivery of all products. This would fulfil producers' expectations and assign UNIPR and the foundation their proper roles.

As a matter of fact, the objective of the two institutions, rural development, entails an implicit weakness of the project to the extent that the underlying principle of inclusive-

ness aims to involve as many farmers as possible, thus preventing an efficient selection of partners.

This also has consequences in other aspects, such as a truly shared vision among farmers and the project outputs. In such a context of heterogeneous behaviours on the part of the farmers, the foundation and UNIPR had to acknowledge that the "wait and see" approach of the majority of the farmers was an obstacle to the progress of the project. Therefore, they decided to implement the project with a small number of farmers, those who proved to be most business-minded and to have a proactive behaviour towards the project activities by putting them into contact with points of sales in the urban market and then to use their commercial results as leverage for later involvement of the remaining farmers. As a matter of fact, at the time being, tangible results have been delivered to some farmers only. This decision clashes with the principle of inclusion but when a project deals with a heterogeneous whole, some of which are poorly motivated beneficiaries, an effective approach can be selecting a small number of proactive participants to be involved in the early implementation of the project, thus achieving two main objectives: on the one hand, preventing the key actors from being frustrated by other participants' behaviour and, on the other hand, gaining the trust of the most hesitating participants by delivering the positive economic effects of the pilot group, acting as a flywheel. In other words, it is the authors' opinion that institutions have to come to terms with the context where the project is embedded and tailor the general principles to the actual potentialities of the community.

However, as much as an incremental implementation can be successful, the main hindering factor remains a widespread lack of trust by the farmers towards institutions that require long-lasting and successful partnerships to be overcome.

What seemed to be most effective for the project was a mutuality to the extent that benefits were not addressed to the farmers only but to the university as well, which benefitted from an excellent opportunity for innovative teaching by bringing the students closer to rural development issues and providing them with an opportunity for on-field learning.

Two tools proved to be very interesting in connecting the students and the farmers for debating real case studies. The first one is a blog on the website where farmers and students interact for problem-solving; the second one is the toolbox used as supporting material for the student competitions within the International Summer School on Food Sustainability organized by UNIPR. In the first two editions of the summer school (in 2020 and 2021), the students engaged in a competition that combined the themes of innovation, resilience, and adaptation of short agri-food chains to the stimuli of the environment in which they operated. Initially thought to be in-person, the summer school changed to online since its first edition due to the pandemic. The competition has proven to act as an excellent tool to cement interactivity among students from all over the world by asking them to present a project that identifies initiatives and strategies at the company, supply chain, and system level for farmers to face current and other potential future shocks. The third edition (June 2022) will be provided in the online mode as well, and again, the student competition will be the core of the school according to which the speakers structure their speeches. This model of interaction between the production and education sectors turned out to be a successful element that is going to be repeated in future editions of the summer school, supporting its sustainability over time.

Thus, the project contributed to the design of tools for participatory teaching that combine theoretical and applied knowledge.

The main facilitating and hindering factors are reported in Tables 3 and 4.

**Table 3.** Facilitating factors of the "Parma, mountains of quality" project.

| Category | Facilitating Factors | Key Findings |
|---|---|---|
| Relational | Trust between partners | The relationship between UNIPR and the foundation was based on mutual trust built up over years of commitment to joint projects. The relationship between the foundation and some farmers ("key farmers") was based on mutual trust, as a result of a collaboration that began in 2018 and was based on shared principles. |
| Relational | Respect among partners | As a consequence of trustful relationships, UNIPR, the foundation, and key farmers honoured and valued one another's opinions. |
| Relational | A good relationship between partners | Because of trustful relationships, UNIPR, the foundation, and key farmers worked well together with strong reciprocal relationships. |
| Relational | Effective and/or frequent communication | UNIPR and the foundation managed to hold good communication, despite the COVID-19 pandemic. |
| Relational | Shared vision, goals, and/or mission | Both the foundation and UNIPR aimed to promote the development of the rural area, thus spending much time reaching as many farmers as possible. |
| Governance | Clearly differentiated roles/functions of partners | The foundation played a facilitating role between the producers and UNIPR, whereas UNIPR supported the Foundation with academic skills and resources. |
| Governance | Good quality of leadership | A UNIPR researcher, holding extra project contacts with key farmers, was appointed to coordinate the project. She acted as a pivot between the students and the farmers. |
| Output | Positive community impact | Tangible results have been delivered to those farmers who had proactive behaviour towards the project activities, by putting them in contact with points of sales in the urban market. |
| Output | Mutual benefit for all partners | Positive outcomes were not addressed to the farmers only but the University as well, which benefitted from an excellent opportunity for innovative teaching by getting the students closer to rural development issues and providing them with an opportunity for on-field learning. |

Source: authors' elaboration.

**Table 4.** Hindering factors of the "Parma, mountains of quality" project.

| Category | Hindering Factors | Key Findings |
|---|---|---|
| Relational | Mistrust among partners | A widespread mistrust was perceived by many farmers towards the university, grounded on several projects that ended up with no consolidated advantages for them. It was found a trustless relationship among some of the farmers themselves. |
| Relational | Partners do not value one another's opinions | Many farmers showed and individualistic behaviour, which is a major problem in the context of marginalized areas. |
| Relational | Poor communication among partners | The pandemic has negatively impacted communication. The kick-off meeting had to be cancelled and telephone instead of face-to-face interviews were conducted. |
| Relational | Lack of shared vision, goals, and/or mission | The foundation and UNIPR aimed to promote the development of the rural area, thus spending much time reaching as many farmers as possible. This was perceived as a project weakness by those farmers who were fast receptive and interested in developing commercial partnerships as quickly as possible. |
| Relational | Differing expectations of partners | The time needed by UNIPR to develop activities and to make resources (skills, human and financial resources, and equipment) available clashed against farmers' expectations of fast solutions. Farmers' expectations were different: some of them proved to be fast receptive, whereas others were poorly motivated and hesitating towards the adoption of new commercial channels. |
| Governance | Unclear roles and/or functions of partners | To fulfil producers' expectations and assign UNIPR and the foundation their proper role, a professional is required to manage those aspects that are out of the UNIPR and foundations functions, such as the products delivery management to urban stores. |
| Governance | Inconsistent partner participation or membership | The majority of the farmers adopted the "wait and see" approach, thus acting as an obstacle to the progress of the project and frustrating the expectations of those farmers who proved to be most business minded. |
| Governance | Excessive time commitment | Many farmers complained about the excessive time commitment in taking part in the survey (i.e., questionnaire filling out). |

Source: authors' elaboration.

The main threat now is that, as the SEU project is over, the UNIPR team's commitment will decline. This is mainly due to the fact there is not yet a clear and full recognition of the third mission activities in Italian universities, thus leading researchers to become mainly involved in research activities. This is a main hindering factor at the root of a vicious circle that prevents the development of permanent and trustful relationships with local communities. In the case study under analysis, at this point of time, the benefits mainly under threat are, on the one hand, the academic marketing skills transfer and

the maintaining/strengthening of the local relational network and, on the other hand, teaching participatory techniques and students' social engagements that come along with the participation in CUPs.

## 5. Conclusions

Despite unforeseen events and difficulties, the third mission, especially in times of uncertainty, can support social groups who suffer a lack of financial or human capital when carrying out specific activities.

For the greatest impact towards sustainability, CUPs need to reap benefits for both the community and the university. In the case of "Parma, mountains of quality", most of the farmers proved to be quite reluctant, aside from a small number of them. To gain farmers' trust, two aspects are crucial, namely the ability to provide stable relationships over time and tangible results.

Indeed, too many times farmers have received support throughout the project life, and at the end of which, institutional participation has declined and so has the sustainability of the project. Such behaviour has reasonably cemented the idea among communities that the university acts as the "House of Wisdom" rather than the "House of Expertise" for societal activities.

It is also likely that if some tangible results are demonstrated, a larger number of farmers might be willing to take part in the project. They may then be available to share some resources to assure the economic sustainability of the project in the long term, such as in the case of "Parma, mountains of quality", where there is a clear need for a professional to manage the logistics of deliveries to the urban market.

The benefits to the universities included the involvement of students who were able to build their soft skills such as empathy and communications skills but also enabled them to apply their learning in a real-world context.

Both stable relationships over time and tangible results require that academic social engagement and the third mission are substantially included in universities' core functions, but full recognition of third mission activities in Italy has yet to come. As the third mission is conceived as a parallel instead of a mainstream activity of the university, it is hard to envisage a change in the methods and contents the university teaches, introducing social research methodology and participatory techniques that emphasise participation and critical thinking. Several alternatives would be available, such as the Community-Engaged Teaching and Learning (CETL) programmes, but these entail a strong top-down commitment on the part of the national Ministry of University.

At the time being, the future of most of the partnerships is currently in question. In a post-COVID-19 world, where societies and economies are struggling to recover and "build back better", students can be seen as agents of change that can create social impact and actively drive the recovery in their region while benefiting from enhanced employability through working on a real-world challenge. By building and enhancing trusting relationships between the universities and their communities, and through creating mutually beneficial opportunities that empower students to make a societal difference, it is hoped that CUPs will leave a legacy beyond European funding.

**Author Contributions:** Introduction, M.G.; Materials and methods, M.C.M. and M.G.; Results, M.C.M. and M.G.; Discussion, M.C.M.; Conclusions, M.C.M.; project administration, M.C.M., M.G. and F.A.; and funding acquisition M.C.M. All authors have read and agreed to the published version of the manuscript.

**Funding:** This research was funded by the "SEU—Socially Engaged Universities" project, co-funded by the Erasmus+ Programme (2014-2020) of the European Union: 2018-1-UK01-KA203-048046.

**Institutional Review Board Statement:** Not applicable.

**Informed Consent Statement:** Informed consent was obtained from all subjects involved in the study.

**Conflicts of Interest:** The authors declare no conflict of interest.

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
