# Peer review of "When Higher Education Meets Sustainable Development of Rural Areas: Lessons Learned from a Community–University Partnership"

_socsci, doi:10.3390/socsci11080338_

Round 1
Reviewer 1 Report
Overall, I found this to be a compelling article, and applaud the commitment required to undertake these multi-layered community-engaged projects during the pandemic.
Specific suggestions are noted below:
Line 7-lower case “institutions” would read better
Line 227 Capitalise “Tables 1 and 2”
Table 2, second row “Bad relationship Partners do not value one another’s opinions. “ is unclear. Is the second part meant to be one of the points in the definition column? Or is it intended to read “Bad relationship as partners do not value one another’s opinions”?
Line 430. Since there was no mention of the “sanitary emergency”, it’s unclear what is being referred to here. Is it the global Covid 19 pandemic?
Line 442 “digital divide” In addition, I am wondering if there is also a lack of trust on the part of the producers about how recordings of digital forms of communication may be used (or may be compromised if it falls into the wrong hands).
Line 450- please indicate how long the questionnaires and interviews took, as it is unclear the extent of time that the farmers found was excessive.
Inclusion of a table that summarizes key findings based on the criteria noted in Tables 1 and 2 that are specific to this case would help more visual readers.
Line 507-I am not familiar with this usage of “in presence”, I am more familiar with “in person”
Line 512-Curious as to why the third edition is in "June 2023" and wasn’t June 2022?
Lines 519-528 - This is more of an aside, but I’m wondering if one of the possible solutions to look into is more sustainable Community-Engaged Teaching and Learning (CETL).
Author Response
Overall, I found this to be a compelling article, and applaud the commitment required to undertake these multi-layered community-engaged projects during the pandemic.
Specific suggestions are noted below:
Line 7-lower case “institutions” would read better
Done
Line 227 Capitalise “Tables 1 and 2”
Done
Table 2, second row “Bad relationship Partners do not value one another’s opinions. “ is unclear. Is the second part meant to be one of the points in the definition column? Or is it intended to read “Bad relationship as partners do not value one another’s opinions”?
The bad relationship stands in not valuing one another’s opinions. We simplified the sentence.
Line 430. Since there was no mention of the “sanitary emergency”, it’s unclear what is being referred to here. Is it the global Covid 19 pandemic?
Yes, it is now specified.
Line 442 “digital divide” In addition, I am wondering if there is also a lack of trust on the part of the producers about how recordings of digital forms of communication may be used (or may be compromised if it falls into the wrong hands).
We haven’t perceived resistance to digital forms of communication, but rather lack of interest in them. We clarified this in the text.
Line 450- please indicate how long the questionnaires and interviews took, as it is unclear the extent of time that the farmers found was excessive.
Done (lines 325-328).
Inclusion of a table that summarizes key findings based on the criteria noted in Tables 1 and 2 that are specific to this case would help more visual readers.
We included Table 3 and Table 4.
Line 507-I am not familiar with this usage of “in presence”, I am more familiar with “in person”
"In presence” has been replaced with in-person.
Line 512-Curious as to why the third edition is in "June 2023" and wasn’t June 2022?
Amended. It was a mistake.
Lines 519-528 - This is more of an aside, but I’m wondering if one of the possible solutions to look into is more sustainable Community-Engaged Teaching and Learning (CETL).
Yes, we agree, it is an option. But it would entail rethinking the whole Master’s Degree course, which is possible, but it would require a strong top-down commitment on the part of the Ministry of University (lines 582-584).

Reviewer 2 Report
Rewrite abstract as follows:
Abstract: Sustainable development in mountainous areas face numerous socio-environmental and economic challenges that public institutions have sought to address for decades. The EU is increasingly demanding Higher Education Institutions be more socially relevant and responsible in addressing the needs of these often underserved communities. To this end, one central principle of the Erasmus+ Project 2014-2020, also consolidated by the Programme 2021-2027, has been to enhance the development, transfer, and implementation of innovative practices fostering social engagement at the local and international level. The paper describes a case study developed in the “Socially Engaged Universities - (SEU)” project, co-funded by the Erasmus+ Programme 2014-2020, which aimed to strengthen the cooperation between universities and local communities. A local Foundation and the University of Parma co-piloted a project to strengthen the short supply chains of a group of farmers in the Italian Apennines. The partnership identified factors that fostered and hampered the effectiveness of community-university partnerships. We propose some final recommendations to ensure that sustainable rural development fully leverages university resources. This includes higher education teaching and research programmes tailored toward the needs of the local community.
*provide more information regarding “the Programme 2021-2027” – is this a EU-wide endeavor?
Line 21: remove facilitating and hindering factors, add (CUP’s) after Community-University Partnership, add Socially Engaged Universities (SEU’s), Erasmus+, social constraints
Line 30: (2022:112) strike :112 not consistent with citation style
Line 35: strike requirement to achieve and replace with prerequisite
Line 42: replace conformed with comprised
Line 43: replace aspects with elements
Lines 66-79: What is the role of Cooperative Extension (or European analogue)? Is this “third mission”? It is outside the traditional teaching and research charge of universities
Line 77: replace coherent with consistent
Line 108: strike In their review and replace with Harney & Willis (2017)
Line 114: strike (Harney & Willis, 2017)
Line 195: strike the , after at al.
Line 219: strike In their paper
Tables 1 and 2: justification and formatting needs to be fixed
Lines 246, 250, 267: complete placeholder XXX’s…
Line 308: discipline of master’s degree students?
Line 420: delete extra space after built up.
Line 531: replace resources with capital
Line 533: replace and with towards
Line 543: replace are provided with demonstrated
Line 544: reword “….project. They may then be available…”
Line 546: reword “…clear need for a professional to manage with logistics of deliveries to the urban market.”
Line 548: reword “The benefits to the universities included the involvement…”
Line 553: reword “As the third…”
Line 568: contributions?
Line 569: No conflict of interest?
General comment: from a curricular standpoint, what’s sorely needed (with regards to first and second mission) is the convergence of rural sociology, ag econ, and agronomy/horticulture, etc. Interdisciplinary, or even transdisciplinary degrees and collaborations that desilo the traditional university departmental structure – and a Cooperative Extension system that can deliver the generated knowledge to stakeholders via a “third mission” mandate.
Line 575, 600-602: See comments for lines 246, 250, and 267
Author Response
Rewrite abstract as follows:
Abstract: Sustainable development in mountainous areas face numerous socio-environmental and economic challenges that public institutions have sought to address for decades. The EU is increasingly demanding Higher Education Institutions be more socially relevant and responsible in addressing the needs of these often underserved communities. To this end, one central principle of the Erasmus+ Project 2014-2020, also consolidated by the Programme 2021-2027, has been to enhance the development, transfer, and implementation of innovative practices fostering social engagement at the local and international level. The paper describes a case study developed in the “Socially Engaged Universities - (SEU)” project, co-funded by the Erasmus+ Programme 2014-2020, which aimed to strengthen the cooperation between universities and local communities. A local Foundation and the University of Parma co-piloted a project to strengthen the short supply chains of a group of farmers in the Italian Apennines. The partnership identified factors that fostered and hampered the effectiveness of community-university partnerships. We propose some final recommendations to ensure that sustainable rural development fully leverages university resources. This includes higher education teaching and research programmes tailored toward the needs of the local community.
Done
*provide more information regarding “the Programme 2021-2027” – is this a EU-wide endeavor?
We added some pieces of information on the Erasmus programme 21-27. Lines 92-96
Line 21: remove facilitating and hindering factors, add (CUP’s) after Community-University Partnership, add Socially Engaged Universities (SEU’s), Erasmus+, social constraints
Done
Line 30: (2022:112) strike :112 not consistent with citation style
Done
Line 35: strike requirement to achieve and replace with prerequisite
Done
Line 42: replace conformed with comprised
Done
Line 43: replace aspects with elements
Done
Lines 66-79: What is the role of Cooperative Extension (or European analogue)? Is this “third mission”? It is outside the traditional teaching and research charge of universities
As now better explained in the text, “third mission”…is “engagement in creating innovative strategies within communities for sustainable development.”… “third mission includes but is not limited to farmers and their communities, being defined as “ all activities of universities that serve a social interest of development in any sphere of civil society” (lines 65-67).
Line 77: replace coherent with consistent
Done
Line 108: strike In their review and replace with Harney & Willis (2017)
Done
Line 114: strike (Harney & Willis, 2017)
Done
Line 195: strike the , after at al.
Done
Line 219: strike In their paper
Done
Tables 1 and 2: justification and formatting needs to be fixed
Done
Lines 246, 250, 267: complete placeholder XXX’s…
Done
Line 308: discipline of master’s degree students?
Done
Line 420: delete extra space after built up.
Done
Line 531: replace resources with capital
Done
Line 533: replace and with towards
Done
Line 543: replace are provided with demonstrated
Done
Line 544: reword “….project. They may then be available…”
Done
Line 546: reword “…clear need for a professional to manage with logistics of deliveries to the urban market.”
Done
Line 548: reword “The benefits to the universities included the involvement…”
Done
Line 553: reword “As the third…”
Done
Line 568: contributions?
Done
Line 569: No conflict of interest?
We specified there are no conflict of interest.
General comment: from a curricular standpoint, what’s sorely needed (with regards to first and second mission) is the convergence of rural sociology, ag econ, and agronomy/horticulture, etc. Interdisciplinary, or even transdisciplinary degrees and collaborations that desilo the traditional university departmental structure – and a Cooperative Extension system that can deliver the generated knowledge to stakeholders via a “third mission” mandate.
We agree. To some extent, this is the aim of the master’s degree in Economics and Management of Sustainable Food Systems at the University of Parma in which there’s a convergence of agricultural and environmental economics, food technology, and rural sociology. We specified it (lines 315-317).
Line 575, 600-602: See comments for lines 246, 250, and 267
Done.
